# Extracellular Vesicles Orchestrate Immune and Tumor Interaction Networks

**DOI:** 10.3390/cancers12123696

**Published:** 2020-12-09

**Authors:** Kevin Ho Wai Yim, Ala’a Al Hrout, Simone Borgoni, Richard Chahwan

**Affiliations:** Institute of Experimental Immunology, University of Zurich, 8057 Zurich, Switzerland; yim@immunology.uzh.ch (K.H.W.Y.); alhrout@immunology.uzh.ch (A.A.H.); simone.borgoni@immunology.uzh.ch (S.B.)

**Keywords:** extracellular vesicles, immune signaling, tumor microenvironment, non-coding RNA

## Abstract

**Simple Summary:**

Significant strides have been made to describe the pervasive role of extracellular vesicles (EVs) in health and disease. This work provides an insightful and unifying mechanistic understanding of EVs in immunity and tumorigenesis. This is achieved by dissecting the role of EVs within the continuum of immune cell physiology, immune–infection responses, and the immune–tumor microenvironment. Our work synthesizes important topical findings on immune EV signaling in mediating immune–tumor interaction networks.

**Abstract:**

Extracellular vesicles (EVs) are emerging as potent and intricate intercellular communication networks. From their first discovery almost forty years ago, several studies have bolstered our understanding of these nano-vesicular structures. EV subpopulations are now characterized by differences in size, surface markers, cargo, and biological effects. Studies have highlighted the importance of EVs in biology and intercellular communication, particularly during immune and tumor interactions. These responses can be equally mediated at the proteomic and epigenomic levels through surface markers or nucleic acid cargo signaling, respectively. Following the exponential growth of EV studies in recent years, we herein synthesize new aspects of the emerging immune–tumor EV-based intercellular communications. We also discuss the potential role of EVs in fundamental immunological processes under physiological conditions, viral infections, and tumorigenic conditions. Finally, we provide insights on the future prospects of immune–tumor EVs and suggest potential avenues for the use of EVs in diagnostics and therapeutics.

## 1. Introduction

Exosomes and microvesicles are two predominant types of phospholipid-bound extracellular vesicles (EVs). Both are released by most eukaryotic cells into the extracellular space, albeit in differing biogenic manners. Smaller EVs—often termed exosomes (30–150 nm) and not to be confused with the unrelated RNA exosome complex [1]—are released via the fusion of multivesicular bodies (MVB) with the plasma membrane. Larger EVs—often termed microvesicles (100–1000 nm)—are released by the outward budding of plasma membrane [2]. Since the nomenclature of EVs is not well standardized in different studies based on their biogenesis mechanisms, to avoid confusion, we will henceforth refer to them as EVs unless specified otherwise. EV cargo in mice and humans is mainly constituted of a combination of RNA (coding and non-coding), proteins, lipids, and possibly DNA (in specific subsets of EVs) that generally reflect the composition of the cell of origin. Research into the intercellular signaling mediated by EVs has revealed the possibility of distal modulation of gene expression, activation, and suppression of immune cells via targeted cargo exchange [3]. As a result, intercellular signaling via EVs has been shown to elicit either positive or negative immune responses, including immune suppression or evasion [4]. Several turning points have highlighted the importance of EVs in biology and intercellular communication, particularly during immune and tumor responses. One of the initial EV reports suggested that B cell EVs express major histocompatibility complex (MHC)-II and can function as antigen-presenting bodies, therefore highlighting their potential in immunity. Subsequent discoveries that EVs can also shuttle RNA cargo between cells, thereby eliciting phenotypic changes in recipient cells, brought further attention to the field through their contribution to epigenomic regulation [4]. To serve as a form of cell–cell communication, EVs carry a wide repertoire of cargo, including proteins, RNA, and metabolites. Cells are selective and precise when sorting and packaging cargo into EVs, often against the gradient of the cell of origin [2]. One mechanism by which EVs can deliver such effects is through the packaging and shuttling of non-coding RNAs (ncRNAs), both small (sncRNA) and long (lncRNA). The non-coding RNA profiles differ greatly from one donor cell to the other, and even between EVs and their parent cells [5], indicating that packaging and shuttling EVs is a precise and controlled process. EV ncRNAs have been shown to be powerful signaling cues that can regulate and modulate the function of recipient cells [2].

## 2. The Multiple Pathways for EV Biogenesis

Several different EV biogenesis mechanisms have been elucidated thus far. However, it is not yet clear if these can account for all EV production mechanisms, and/or if some of these mechanisms could act redundantly together [6]. Nonetheless, these mechanisms could be broadly classified as endosomal sorting complexes required for transport (ESCRT)-dependent or -independent, and are further illustrated as follows.

The endocytic pathway (i.e., ESCRT-dependent) is characterized by highly dynamic membrane compartments involved in the internalization of extracellular ligands or cellular components, their recycling to the plasma membrane, and/or their degradation [7]. Early endosomes mature and form late endosomes, in which intraluminal vesicles (ILVs) are accumulated in the lumen. These late endosomes filled with multiple ILVs are generally referred to as multivesicular bodies (MVBs) and consist of proteins, lipids, and cytosol that are specifically sorted. The more common fate of MVBs is fusion with lysosomes that contain lysosomal hydrolases to degrade their cargo [8]. However, organelles with MVB markers such as tetraspanins (CD9, CD63 and CD81), lysosomal-associated membrane proteins (LAMP1 and LAMP2), and other molecules associated with late endosomes (i.e., MHC-II in antigen-presenting cells), can also fuse with the plasma membrane, releasing their content into the extracellular matrix [9]. The formation of MVBs and ILVs is mainly driven by the endosomal sorting complex required for transport (ESCRT), which consists of four complexes (ESCRT-0, -I, -II and -III) with associated proteins (VPS4, VTA1 and ALIX), conserved from yeast to mammals. The ESCRT-0 complex binds and clusters ubiquitinated transmembrane proteins in the endosomal membrane. ESCRT-I and -II complexes are responsible for membrane remodeling into buds and the delivery of ubiquitinated protein cargo, and ESCRT-III components subsequently facilitate vesicle abscission. ESCRT-0 consists of a hepatocyte growth factor–regulated tyrosine kinase substrate (HRS) that recognizes ubiquitinated proteins and associates with the signal transducing adaptor molecule (STAM). HRS further recruits TSG101 from the ESCRT-I complex to the endosome, and ESCRT-I is then involved in the recruitment of ESCRT-III via ESCRT-II or ALIX. Finally, the dissociation and recycling of the ESCRT machinery is promoted by the interaction with the AAA-ATPase VPS4 that triggers the disassembly of the ESCRT-III complex [6].

The ESCRT-independent formation of ILVs and MVBs [10] has two variants [11]: (i) tetraspanin-dependent EV formation, whereby tetraspanin factors such as CD63 and CD81 are shown to be functional in the formation of ILVs (<40 nm), and sorting target receptors and intracellular components into EVs independently of the HRS and ESCRT components [12]. Reduced secretion of EVs, as measured by nanoparticle tracking analysis, was shown to be the result of CRISPR/Cas9 knockdown of CD63 in HEK293 cells [13]. (ii) Ceramide-dependent EV formation, which is independent of ESCRT components, involves the inhibition of neutral sphingomyelinase activity (responsible for ceramide synthesis) by GW4869. These EVs are released primarily by HEK293 cells, macrophages and other tumor cell lines [6]. Common EV markers (CD63, CD81 or TSG101) and miRNA secretion were decreased upon GW4869 treatment in HEK293 cells [14]. Indeed, the dependence on ceramide in EV production was shown to vary across different cell types. GW4869 treatment in human melanoma cells does not affect MVB biogenesis, nor EV secretion [15]. In primary cells, GW4869 treatment induces significant cell death, leading to the unreliable analysis of EV secretion [6].

## 3. The Role of EVs in Immune Homeostasis

An increasing number of studies have shown that EVs are constitutively secreted by most immune cell types, such as B and T cells, natural killer cells, dendritic cells, and macrophages. Considering the intrinsic nature of EV biogenesis, which involves cytoplasmic endosomal sorting, it is not surprising that immune cell-derived EVs carry proteins and nucleic acids from the cell of origin within the lumen, as well as cell-specific antigens on the surface, reflecting their developmental status [3]. EV surface antigens allow for the direct stimulation of recipient cells, as well as targeted homing to specific cell types after release from parental cells. Molecular cargo is well conserved in the lipid bilayer of enclosed EVs, allowing stable and high dimensional (i.e., RNA, DNA, protein, lipid) delivery from one cell to another for potent gene regulation in recipient cells. Taken together, these features allow EVs to influence host immune responses and provide an additional form of intercellular regulation, which is likely to be more complex and intricate than conventional cytokines or antigen stimulation (Figure 1).

### 3.1. The Contribution of EVs in Humoral Immunity

The activation of murine B cells via interleukin-4 (IL-4) and CD40 receptors induces a high level release of EVs. However, CD63 (a conventional EV marker) is not detectable in murine B cell EVs, but rather in the cell of origin [16], suggesting that CD63 is not a robust EV marker for studying B cell-derived EVs in mice. An enriched amount of MHC-I/-II peptides, B cell receptors (BCR), and other costimulatory molecules like CD54 and CD86 were found in both human and murine B cell-derived EVs [3]. The B cell EV-mediated delivery of MHC-II has been shown to activate T cell immune responses by increasing proliferation and the production of T-helper (T_H_)2-like cytokines [17]. Additionally, they have also been shown to induce the cytotoxic activity of CD8^+^ T cells in vivo independent of host BCR expression and circulating antibodies [18]. The localization of MHC-II from B cell EVs is also vital for follicular dendritic cells, which are unable to synthesize MHC-II themselves to activate other immune cells in an MHC-II-dependent manner [19,20]. The expression of adhesion molecules like ICAM-1, α4, and β1 and β2 integrins is pronounced in B cell EVs. Integrins on EVs are functional in the context of adhesive interactions between EVs, extracellular matrix components and other cells in the environment during inflammation [21]. EVs derived from a Burkitt’s lymphoma cell line (DG75) induce healthy human IgD+ primary B cells to proliferate and upregulate the expression of activation-induced cytidine deaminase and germline transcripts for IgG1, indicating the ability of EVs to induce class-switch recombination (CSR) and to drive B cell maturation [22,23,24]. Recently, B cell-derived EVs were utilized to deliver exogenous ncRNA cargo to other immune cells in vivo. An exogenous mir-155 inhibitor or mimic was specifically packaged into B cell EVs and shown to modulate immune responses in macrophages and hepatocytes [25]. Whilst these are exciting developments in B cell physiology, evidence is still lacking as to whether B cells can communicate among themselves through EVs (Figure 1).

### 3.2. The Contribution of EVs in T Cell-Mediated Immunity

Activated T cells release EVs that have been shown to regulate various immunological responses in dendritic cells, B cells, and exogenous T cells [26]. Unlike B cell EVs, conventional tetraspanin EV markers, including CD63, are expressed more in T cell EVs than their parental cells [27]. While CD3 was readily detectable in activated CD3^+^ T cell EVs, no expression of MHC-I/-II peptides, CD4, CD25, CD40 nor ICAM-1 was detected. The addition of activated CD3^+^ T cell EVs to resting autologous CD3^+^ T cells stimulated with IL-2 displayed altered responses compared to stimulation with IL-2 alone, resulting in increased proliferation and changes in cytokines production [28]. Human regulatory T cells (Tregs) can produce EVs to inhibit the proliferation of effector T cells (Teffs) in a dose-dependent fashion. This inhibition generates a switch in the cytokine profile of Teffs, with increased production of IL-4 and IL-10 and decreased production of IL-6, IL-2 and interferon (IFN)-γ secretion [29]. Similarly, activated CD4^+^ T cell-derived EVs were found to have an enriched amount of tRNA fragments (tRFs) compared to their resting naïve state, in accordance with the fact that tRFs can inhibit T cell activation. This form of selective tRF removal via EVs promotes T cell activation in physiological conditions [30]. Murine CD4^+^ T cell-derived EVs can influence B cell immune responses in vivo by enhancing proliferation, germinal center reactions, and antibody production via transferring specific sets of miRNAs or the binding of CD40L to CD40 [31]. Taken together, these results suggest that EVs are a new class of regulators in T cell immunity via the delivery of sets of functionally active nucleic acids and proteins, which are possibly responsible for modulating autoimmunity or immunodeficient diseases. However, current EV studies are mostly based on total EV populations derived from one cell type, which are relatively heterogeneous in terms of size, molecule expression and cargo. Thus, future studies should attempt to study different EV subsets, as the molecular profile within each subset could be very different, and possibly dilute the phenotypic and genotypic effect (Figure 1).

### 3.3. The Contribution of Natural Killer Cell-Derived EVs

Natural killer (NK) cells also play a vital role in innate immunity by providing cytotoxicity independently of cytokines or MHC signals. Recent studies have also shown their ability to regulate adaptive immune responses via modulation of the immune environment [32]. EVs are released by resting and activated NK cells, and the exosomal marker CD63 and the parental marker NKG2D are readily detected on their surface [33]. Studies have shown that both resting and activated human peripheral NK-derived EVs can exert cytotoxic activity on activated but not resting immune cells, as well as on several cancer cell lines derived from T cell leukemia, erythroleukemia, Burkitt lymphoma, and metastatic breast adenocarcinoma [32]. This suggests that EVs are one of the tools used by NK cells to potentially increase their coverage of immune surveillance. However, the contribution of NK cell EVs in the context of immune regulation and interaction with other immune cells is still very understudied. Indeed, future studies on their functional roles would enhance our understanding of EV-mediated NK cell immune regulation and could usher a new era of innate–adaptive immune interactions via EVs.

### 3.4. The Contribution of EVs in the Immune Response of Antigen-Presenting Cells

Professional APCs, such as dendritic cells (DCs) and macrophages, also produce EVs to modulate immune responses in other immune cell types. EV-specific proteins, such as ESCRT proteins, tetraspanins, cytoskeleton proteins and heat shock proteins, are expressed readily on DC-derived EVs [34], suggesting that related pathways might be involved in the biogenesis of DC EVs. DC EVs also carry abundant amounts of cell type-specific markers like MHC-I and -II. Interestingly, MHC-II molecules are stored within luminal vesicles in immature DCs. During activation, the MHC-II-containing luminal vesicles fuse with MVBs, followed by fusion with the cell’s plasma membrane, which subsequently increases the amount of surface MHC-II for antigen presentation [35]. Moreover, immunomodulatory mir-146a and mir-155 were found in bone marrow-derived DC (BMDC) EVs in mice. These two miRNAs have been demonstrated to be transferred between BMDCs, with EV-delivered mir-146a suppressing inflammatory responses to lipopolysaccharide (LPS) in recipient wild type (WT) mice. However, mir-155 transferred between BMDCs via EVs promoted inflammatory responses to LPS in recipient mir-155 knockout mice [36], suggesting that EVs can exert both positive and negative effects on host immunity. If the enrichment of mir-146a or mir-155 occurs in different EV subpopulations, one could purify certain EV subpopulations from the same parental cells to serve different biological and clinical purposes. Macrophage-derived EVs also express tetraspanins and heat shock proteins similarly to other cell types, although the expression levels are not significantly enriched compared to cell lysates. EVs released from pathogen-infected (mycobacteria, salmonella, or toxoplasma) macrophages carry MHC-I and -II, as well as costimulatory molecules such as CD86 that are able to activate CD4^+^ and CD8^+^ T cell responses both in vitro and in vivo [37]. Overall, EVs are becoming a novel and legitimate class of intercellular signaling players between immune cells to orchestrate immune responses in physiological conditions, pathogenic infections, and cancer development in mammalian systems. However, the field of EV studies is still in early development. The identification and functional studies of significant EV cargo in different immune contexts are contributing multidimensionally to the field by filling some missing gaps in certain known pathways. Combining this knowledge and introducing this new viewpoint for studying immune regulation will provide novel mechanisms that could be strategically implemented into clinical diagnostics and therapeutics.

## 4. The Emerging Role of EVs in Viral–Immune Interactions

Host–pathogen interactions can drastically affect immune signaling [38]. This is most obvious with viruses, which are obligate intracellular pathogens. Under these conditions, EVs have been shown to: (i) act as viral decoys for immune recognition [39]; (ii) increase the breadth of viral virulence [40,41]; and (iii) potentially increase viral tropism [42,43] (Figure 2). A fourth hypothetical scenario we propose here is the potential for viral EVs to act as pseudo-morphogen gradients for both immune and non-immune cell recruitment. While EVs have been suggested to create a pseudo-morphogen gradient in the past, it remains to be shown whether viral EVs could mediate a similar response (Figure 2).

It has been shown that viruses are able to induce EV secretion in infected cells, and these EVs can mediate the spread of different viruses. One example is Coxsackievirus B1, which through the depolymerization of the actin cytoskeleton of the infected cell, induces the release of EVs and enables virus spread [44]. Other non-enveloped viruses, such as picornaviruses and enteroviruses, have been described to use EVs to spread in the host [45,46]. Surprisingly, this is also true for infection between species. Indeed, it was recently reported that the transmission of Langat virus from arthropods to humans is mediated by EVs that are secreted by ticks, which deliver viral RNA and proteins to human keratinocytes [47]. At the same time, EV-mediated communication is also used by the host as a defense mechanism against viral infection, activating the immune response in non-infected cells. The role of EVs in viral infection has been characterized for human immunodeficiency virus type 1 (HIV) and the Epstein–Barr virus (EBV). In HIV, EVs can both positively and negatively promote tumor spread [48,49]. For example, CD4 has been shown to be essential for HIV entry to specific cell types such as M1 macrophages and fibroblasts, and it was recently shown that T cell EVs expressing CD4 on their surface could act as a positive decoy for the virus, thereby acting as bait to recruit the virus away from the cells and avoid cell infection [39,49]. At the same time, CD8+ T cell-derived EVs express surface factors that have proven antiviral activity. On the other hand, EVs have been shown to enhance HIV infection through several mechanisms. An example of this is the EV-based transfer of CCR5 and CXCR4 receptors to other cells, rendering them susceptible to HIV entry [50,51]. Additionally, EVs can also transfer viral proteins such as Env, Gag, and Nef that could help viral spread and promote immunodeficiency [52]. As supporting evidence of the central role of EVs in HIV infection, it has been shown that plasma from HIV patients contains higher levels of EVs and that these EVs contain more pro-inflammatory cargo [53,54] (Figure 2).

Numerous recent studies have shown that Esptein–Barr virus (EBV), a human gamma herpesvirus associated with lymphoproliferative disorders such as Burkitt lymphoma (BL) and diffuse large B-cell lymphoma (DLBCL), utilizes EVs as a signaling tool to facilitate viral transmission via modulation of host immune defense [55]. During DLBCL formation in a humanized mouse model, EVs derived from EBV transformed lymphoblastoid cell lines (LCLs) and switched macrophages to a proinflammatory state via the delivery of EBV miRNAs. Strikingly, the injection of EVs derived from one of the LCL EBV^+^ strains significantly reduced the survival rate of the mice [43]. EVs derived from LCLs have also been shown to be taken up by DCs, where the EBV miRNA is delivered. This transfer of EBV miRNA from LCLs to DCs led to the repression of *CXCL11* (an immunoregulatory EBV target gene) reporter expression in recipient DCs [56]. Latent membrane protein 1 (LMP1), an EBV specific protein, has been shown to be expressed on the surface of EBV-infected cell-derived EVs. LMP1 on these EVs possibly inhibits T cell proliferation and NK cell toxicity [22,57]. EBV-infected cell-derived EVs also express galectin-9, which induces the apoptosis of EBV-specific CD4^+^ T cells through an interaction with T cell immunoglobulins [57]. A paracrine loop of EBV lytic replication enhancement has recently been shown to be an EV-mediated process. EBV-infected B cells secrete EBV-encoded non-coding RNA-containing EVs to neighboring infected B cells, where the EVs increase *CXCL8* expression via endosomal Toll-like receptor (TLR) 7, ultimately promoting lytic replication, which is vital for viral propagation [42]. Taken together, EVs are used by viruses as a novel tool to promote their propagation through modifications to immune responses in the host, thus increasing viral tropism. However, compared to professional APCs, little is known about the interactions between EBV-infected cell-derived EVs and non-infected lymphocytes. One study has shown the presence of EBV miRNAs in non-infected human peripheral B cells, suggesting the possibility of EV-mediated EV transfer from EBV infected cells to other B cells (Figure 2).

## 5. EV-Mediated Crosstalk in Immune–Tumor Cell Interactions

Rapidly cell division represents only one factor in the complex process of tumorigenesis. Tumor surroundings are composed of a dynamic network of non-malignant cells, non-cellular components, signaling molecules, and extracellular matrices (ECM) [58,59,60] which collectively form the tumor microenvironment (TME). The TME is involved in a bi-directional interaction with the tumor mass to sustain and contribute to the growth and spread of the tumor [60]. Such communication is underlined in an increasing body of evidence that highlights the key role played by the TME in tumor progression [59,61,62,63]. In addition, many studies have reported the positive role of the TME in restraining tumor initiation and progression at initial stages of carcinogenesis [64], and how “re-programming” the TME in later stages holds great potential in terms of effective cancer treatment [59] (Figure 3).

### 5.1. Tumor–Tumor Interactions via EVs

Cancer cells communicate with each other to sustain the growth and survival of the tumor in a hostile microenvironment. The release and uptake of EVs containing signaling cargo is one form of cell–cell communication used by cancer cells. EVs have been implicated in cancer progression by promoting proliferation, angiogenesis, and immune suppression. Within the primary tumor itself, autocrine signaling, mediated through the shuttling of EVs, promotes a more transformed phenotype in the recipient cells. In glioma cells, EVs were shown to shuttle oncogenic proteins to recipient cells, therefore leading to altered signaling and proliferation despite the lack of genomic alterations. For example, the transfer of epidermal growth factor receptor (EGFR) variant III (EGFRvIII) from expressing glioma cells to non-expressing cells aided their transformation and enhanced cancer growth [65]. Similarly, EVs have been shown to promote cancer proliferation and progression in chronic myeloid leukemia [66], brain [67,68], gastric [69,70], and bladder [71] cancers. Cancer autocrine signaling via EVs is also involved in other pro-tumor processes, including conferring drug resistance in sensitive cells, which will be discussed in more detail in the following sections.

### 5.2. Tumor–Immune TME Interactions via EVs

One critical component of the TME are immune cells, which are involved in a cross-talk with the tumor via several secreted proteins and EVs. Cancer cells are able to recruit immune cells to promote the growth and survival of the tumor (Figure 3).

The important role of T cells in antitumor immunity is well established, as they are considered the major contributor to antitumor immunity, being the most common tumor-infiltrating lymphocytes in the tumor vicinity. Studies have reported on the immune suppressive effects of cancer-derived EVs on T cells, either by inhibiting their proliferation [72], inducing apoptosis [73], or diminishing their cytotoxic ability [74]. For instance, melanoma EVs are shown to target CD4^+^ T cells to promote the growth of the tumor by inducing caspase-mediated apoptosis once up taken by T cells [75]. These findings are consistent with data showing that melanoma-derived EVs isolated from plasma significantly induce apoptosis and inhibit proliferation in CD8^+^ T cells [76]. Similarly, EVs of breast, lung, and nasopharyngeal carcinomas have been implicated in targeting T cells by inhibiting their proliferation and blocking their differentiation and activation [77,78]. Moreover, melanoma cell line-derived EVs were shown to influence the epigenetic landscape of recipient cytotoxic T cells via the delivery of a subset of non-coding and coding RNAs. This supports the the potential of EVs as a new class of epigentic regulator [79].

B cells are central to humoral responses and antitumor immunity. In addition to producing antibodies against a limitless array of targets, B cells are able to shape the functions of other immune cells by antigen presentation, co-stimulation, and cytokine secretion [80,81,82]. Indeed, tumor-infiltrating B cells have been reported to be involved in different stages of carcinogenesis, shaping the immune response according to microenvironmental cues [82,83]. For example, mycoplasma-infected melanoma cells release EVs that selectively activate B cells to secrete IL-10, leading to the inhibition of T cell proliferation and activity [84]. Restricting the exposure of B cells to melanoma-derived EVs by CD169^+^ macrophages has been shown to block tumor-promoting humoral immunity and suppress tumor growth [85]. Similarly, B cells exert an immunosuppressive effect in response to esophageal cancer-derived EVs. Naïve B cells differentiate into regulatory B cells after exposure to esophageal cancer-derived EVs, where they produce TGF-β, leading to CD8^+^ T cell inhibition [86].

Another major contributor in the immune TME are NK cells, which participate in immunosurveillance and tumor suppression. Several studies have reported on the role of NK cell-derived EVs in different cancers, and their antitumor effects in vitro and in vivo, even under immunosuppressive conditions [87,88]. It has been shown that NK cell-derived EVs can activate different cell death pathways in cancer cells, including endoplasmic reticulum ER stress-induced apoptosis, and caspase-dependent and -independent pathways [89]. NK cell-derived EVs display the DNAM1 receptor on their surface, which enhances their binding and induces cytotoxicity towards target cancer cells [90]. The cytotoxic activity of NK cell-derived EVs has also been shown to be exerted at low concentrations and after a short time exposure [90]. This observation is consistent with the activating role of DNAM1 in the anti-tumor response of NK cells [91,92], where it also plays a key role in immunosurveillance [93,94,95,96] and the suppression of metastasis [97].

In addition, NK cell-derived EVs have been shown to shuttle certain miRNAs that induce an anti-tumor response even in immunosuppressive TMEs [98], where they can exert cytotoxic effects specifically against tumor cells [32]. Indeed, the newly studied miR-3607-3p is found to be enriched in the EVs of NK cells as a way to deliver them to pancreatic cancer cells [99]. NK cell- and EV-derived miR-3607-3p inhibited the proliferation, migration and invasion of pancreatic cancer cells through the inhibition of IL-26 [99]. These findings are consistent with the suppressive role of miR-3607-3p that has been reported in non-small cell lung cancer (NSCLC), where it has been shown to induce cell cycle arrest and inhibit tumor growth and metastasis [100]. Interestingly, miR-3607-3p has also been reported to be downregulated in both NSCLC and pancreatic cancer tissues, and its expression can be used as a prognostic predictor for survival [99,100]. In neuroblastoma, NK cell- and EV-derived miR-186 plays a similar role in suppressing cancer cell growth and preventing TGFβ1-dependent immune escape [89]. miR-186 has further been shown to regulate many processes in different human cancers, and to be a marker for diagnosis and prognosis [101].

On the other hand, tumor cells can release EVs to limit the cytotoxic capacity of NK cells. Co-culturing NK cells with immunosuppressive pancreatic cancer-derived EVs expressing TGF-β1 resulted in the significant downregulation of NKG2D, CD107a, TNF-α, and INF-γ in NK cells, reducing their cytotoxicity against pancreatic tumors [102]. Similarly, membrane-associated TGF-β1 present on EVs derived from the sera of acute myeloid leukemia patients inhibited natural killer cell activity and downregulated NKG2D [103]. Also, hypoxic tumor-derived EVs have been shown to inhibit NK cell cytotoxicity in vitro and in vivo through shuttling of TGF-β and miR23a [74,104]. In addition, treatment with tumor-derived EVs decreased NK cell percentage in lungs and spleens in a murine mammary carcinoma model. EVs derived from mammary carcinoma suppress NK cell function in vitro and ex vivo [101].

### 5.3. Tumor–Stroma Interactions via EVs

The tumor stroma has been shown to be involved in the process of tumorgenesis in early stages by providing either a supportive or inhibitive “soil” for growth. The components of the stroma (i.e., fibroblasts, mesenchymal stem cells, and the ECM) have been shown to uptake cancer derived-EVs and secrete EVs containing miRNA/lncRNA/protein cargo to enhance tumor growth and survival. Mesenchymal stem cell (MSC)-derived EVs promoted tumor growth in different cancers both in vivo and in vitro via inducing VEGF overexpression and the activation of the ERK1/2 pathway [105]. In breast cancer, MSC-derived EVs were shown to secrete mir-222/223-containing exosomes to stimulate cancer cell quiescence, and subsequently drug resistance [106]. Similarly, MSC-derived EVs increased the expression of multidrug resistance-associated protein (MDR) proteins and activated CaM-Ks/Raf/MEK/ERK pathways to promote drug resistance in gastric cancer cells [107]. Fibroblast-derived EVs promoted the growth and metastasis of colorectal cancer in both in vitro and in vivo models [108], and promoted a more aggressive phenotype in breast and pancreatic cancer cells [109,110]. Fibroblast-derived exosomal miR-21, -106b, -98, and -522 have been implicated in driving drug resistance in ovarian, gastric, and pancreatic cancers [111,112,113,114]. Stromal-derived EVs also play a role in other hallmarks of cancer drivers including migration and metastasis [115,116,117,118].

### 5.4. Cancer Stem Cell Signaling via EVs

Cancer stem cells (CSCs) are key players in the tumor microenvironment, where they have been highlighted as major contributors to drug resistance. CSCs and other cancer cells are in a continuous state of equilibrium where non-CSCs can de-differentiate into CSCs, and vice versa. Several studies have implicated exosomes as carriers of de-differentiation cues. Exosomal-mediated Wnt signaling has been shown to drive the de-differentiation of cancer cells to obtain stem-like properties and phenotypes in diffuse large B-cell lymphoma [119] and colorectal cancer [120,121]. Stroma-derived exosomes shuttling IL-6 and activin-A were shown to de-differentiate lung cancer cells into CSCs, and promote the activation of stemness-promoting pathways including the Wnt pathway [122]. Colorectal cancer stem cell-derived exosomes promoted stem-like properties and phenotype in colon cancer cells via miR-146a, in addition to enhancing their tumorigenic and immunosuppressive abilities [123].

## 6. Mechanisms of Immune–Tumor Cell Communications

EVs have been implicated in cancer progression by promoting proliferation, angiogenesis, and immune suppression [124]. Immune cells are recruited to the tumor vicinity as a response to TME cues, an example of which is EV-mediated signaling, whether it be immune-activating or immune-suppressing (Figure 3).

### 6.1. Uptake-Dependent Communication via sncRNA

Cancer cell- and EV-derived miRNAs have been implicated in several cancer-promoting pathways. Many studies have shown the role of EV miRNAs in promoting angiogenesis, a key hallmark of cancer progression, in several cancers. For example, colorectal cancer cells package miR-1229 into their EVs to target the HIPK2/VEGF pathway, promoting angiogenesis in vitro and in vivo [125]. Treatment with an antagomir against miR-1229 impaired tubulogenesis of HUVECs and inhibited tumor growth and angiogenesis in xenograft models [125]. Moreover, miR-210 has been implicated in promoting angiogenesis and enhancing tube formation in endothelial cells co-cultured with hypoxic leukemia cell EVs [126]. In addition, miR-210 has been shown to promote angiogenesis in lung adenocarcinoma and hepatocellular carcinoma (HCC). Overexpressing *TIMP-1* led to the upregulation of miR-210 in lung adenocarcinoma-derived EVs in vitro and in vivo, subsequently leading to enhanced tube formation in HUVECs and increased angiogenesis in xenograft models [127]. Similarly, HCC cells package miR-210 in their EVs to promote tubulogenesis in vitro and enhance angiogenesis in xenograft models via the SMAD4 and STAT6 pathways [128].

EVs are also excellent mediators of metastasis as they can exert their effects in local or remote microenvironments, as well as in an organ-specific manner [129]. EV-based communication has been implicated in preparing a hospitable pre-metastatic niche and promoting tumor spread in several cancers by different means, such as localizing in common sites of metastasis and increasing endothelial permeability [130], helping in the formation of a pro-inflammatory niche [131], and destroying the blood–brain barrier to promote brain metastasis [132].

It has become clear that tumors can transmit pro-metastatic cues to their TME via miRNA packaged in their EVs. In triple-negative breast cancer cells, miR-939 and miR-105 are packaged in EVs, where they target vascular endothelial cadherin (*VE-cadherin*) and Zonula occludens-1 (*ZO-1*), respectively, once up taken by endothelial cells, resulting in the disruption of the endothelial barrier and increased permeability [133,134]. Exosomal miR-103 plays a similar and critical role in the metastatic process of HCC via directly targeting multiple endothelial junction proteins of endothelial cells, thus increasing vascular permeability and allowing metastatic formation [135].

Multiple studies have reported on the role that miR-10b plays in metastasis formation. It has been implicated in the progression of 18 cancer types including, but not limited to, breast, brain, and gastrointestinal tumors [136]. More recently, studies have directed their focus on the part that EVs play in miR-10b-mediated metastasis formation. In HCC, miR-10b promotes migration, invasion, and tumor spreading via the inhibition of CADM1 [137,138], a suppressor of invasion and survival, and through modulating matrix metallopeptidases expression [123], which are key participants in ECM remodeling and metastasis. Notably, the overexpression of miR-10b from HCC patient sera EVs can even be associated with poor disease-free survival [139]. These findings are consistent with others showing a considerable increase in miR-10b in HCC EVs, leading to enhanced proliferation, migration, and invasion in vivo and in vitro [140]. miR-10b level was elevated in EVs derived from the sera of early HCC patients, where it was linked to advanced tumor stage and an independent prognostic factor for early HCC patient disease-free survival (Figure 3).

Interestingly, cancer- and EV-derived miRNAs can also promote the metastatic process indirectly through “reprogramming” of the TME to become complicit in driving tumor spread. Exosomal miR-10b secreted by CRC cells modulates fibroblasts in the surrounding stroma via suppression of PIK3CA expression and downregulation of the PI3K/Akt pathway [141]. Once taken up by fibroblasts, EVs containing miR-10b transform the cells into cancer-associated fibroblasts expressing myofibroblastic markers that are capable of promoting CRC growth in vitro and in vivo. A similar “reprogramming” of fibroblasts to promote the invasion and formation of a pre-metastatic niche has been reported in triple-negative breast cancer and melanoma via EV-derived miR-9 [142] and miR-155/miR-210 [143], respectively.

### 6.2. Uptake-Dependent Communication via lncRNA

EVs are also known to package other lncRNAs to exert a response in recipient cells. EV-derived lncRNAs have been implicated in many aspects of cancer progression. In NSCLC patients, metastasis-associated lung adenocarcinoma transcript 1 (MALAT-1) has been shown to be highly expressed in the serum, and that the level of exosomal MALAT-1 correlates with tumor stage and metastasis status [144]. Moreover, exosomal MALAT-1 seemed to play a supporting role in migration and proliferation. Another lncRNA involved in EV-mediated metastatic spread is Hox antisense intergenic RNA (HOTAIR). HOTAIR has been shown to be packaged in EVs from laryngeal squamous cell carcinoma [145] and urothelial bladder cancer [146] cells, promoting metastasis formation and correlating with clinical stage. EV-derived HOTAIR has also been shown to play a role in angiogenesis in neuroblastoma, where glioma cells shuttle it to endothelial cells to induce Vascular Endothelial Growth Factor A (VEGFA) expression [147] and silencing of HOTAIR-inhibited glioma-induced endothelial cell proliferation, migration, and tube formation [147]. EV-based lncRNA is also implicated in promoting therapy resistance. For instance, the high expression of urothelial cancer associated 1 (UCA1) in breast cancer EVs correlates with tamoxifen resistance [148]. In renal cell carcinoma, sunitinib-resistant cells highly express lncARSR, which in turn is packaged in RCC EVs to be shuttled to recipient cells [149].

### 6.3. Uptake-Independent Communication

Most studies have shown that EV-mediated communication is cargo uptake dependent, in which EVs release their cargo after being taken up by recipient cells. However, there is emerging evidence that shows the ability of EVs to deliver signals independent of uptake via surface proteins. Displaying proteins on the surface of their EVs allows cancer cells to exert immunosuppression systemically.

Malignant ascites from ovarian cancer patients release soluble E-cadherin on the surface of EVs, promoting angiogenesis via the activation of β-catenin and NFκB signaling in endothelial cells [150]. Similarly, EVs derived from ovarian, colorectal, and renal cancer cells express the 189 amino acid VEGF isoform on their surface [151]. VEGF_189_ EVs were shown to promote endothelial cell migration and tube formation, in addition to angiogenesis and xenograft tumor growth, independently of uptake. Interestingly, the VEGF_189_ isoform was more favorably localized to cancer-derived EVs than other isoforms of VEGF via high affinity binding to heparin, increasing its half-life and impairing its recognition by the VEGF antibody bevacizumab.

### 6.4. EV-Mediated Tumor Resistance to Immunotherapy

EVs have emerged as a potent mode of drug/therapy resistance in cancer, where they have been shown to enhance resistance in aggressive cancers and even to transfer resistance to cancer cells that were once sensitive to treatment. One way in which resistant cancer cells counter treatment is through encapsulating anti-cancer agents into EVs and expelling them into the extracellular environment, which has been reported in several cancers. Indeed, breast cancer cells have been shown to package doxorubicin in their EVs to eject it outside the cell [152]. Similarly, a number of studies reported that melanoma [153], ovarian [154], and lung cancer cells [155] encapsulate cisplatin in their secreted EVs to eject it into the extracellular environment. Interestingly, glioblastoma EVs are enriched in autologous neoantigens that are capable of inducing DC-mediated anti-tumor immune responses [156], thus revealing a new class of cell-free presentation of neoantigens for anti-tumor immunity.

Melanoma immunotherapy resistance depends on the overexpression of PD-L1, and it has been recently shown that the majority of melanoma PD-L1 is shuttled through EVs and not displayed on the cell surface [157,158,159]. EV-derived PD-L1 has been shown to elicit immunosuppressive effects similar to cellular PD-L1, which promotes immune evasion in melanoma tumors by inhibiting CD8^+^ T cell responses [160], where the blockade of PD-L1 induces systemic anti-tumor immunity [157,158]. Similarly, aggressive B cell lymphomas are able to evade immunotherapy targeting CD20, such as rituximab, by releasing EVs with CD20 on their surface to shuttle the antibodies away from the lymphoma cells [161].

EVs can also transmit resistance by transferring bioactive cargo such as proteins, miRNAs, and lncRNAs from resistant to sensitive cancer cells. For example, EVs are thought to confer drug resistance in osteosarcoma [162], where cancer cells with multi-drug resistance (MDR) transferred the drug resistance phenotype to sensitive cells in vitro by packaging MDR proteins into their EVs. This allows recipient cells to expel uptaken drugs and to stop the accumulation of anti-cancer drugs inside the cell. Similar findings have been reported in acute T lymphoblastic leukemia [163] and prostate [164] and ovarian [165] cancers. Moreover, EV-transferred resistance via P-glycoprotein 1 (P-gp1) could be permanent, or at least prolonged, both in vitro and in vivo in many tumor types [166]. This observation could be partially explained by the EV-derived miRNA-mediated regulation of P-gp expression by miR-27a and miR-451 [167,168]. EV-packaged miRNAs have also been shown to promote resistance through modulating the expression of different genes. In breast cancer cells, miR-222, -221, -100, -30a, -17, -23a, and -149 were shown to be taken up into drug-sensitive cells via EVs derived from resistant cells, which promoted resistance against a number of drugs including docetaxel, tamoxifen, and adriamycin [169,170,171,172].

## 7. Future Therapeutic Perspectives for EVs

The ability of EVs to mediate intercellular communication and modulate target cell behavior has several potential therapeutic applications. EVs have been proven to be optimal targets for early detection in liquid biopsy of several cancer entities [173]. Due to technical limitations of the current cytometers, as well as the heterogenity of EVs derived from patients, the implementation of successful EV-based diagnosis has been delayed. Innovative technologies for EV studies have emerged in the past few years, such as super-resolution microscopy and multiplex imaging cytometry. However, the rather low throughput and tedious sample preparation of these technologies are still significant barriers for robust basic and clinical EV research. Recent developments in nanosize particle-specialized cytometers and optimized materials for calibration beads are set to overcome the issues of sensitivity and unspecificity in current instruments [173]. This advancement will ultimately allow for the reproducible measurement of single EV particles and their surface markers in a high throughput manner. To tackle the inconsistency and reproducibility from one study to another, Minimal Information for Studies of Extracellular Vesicles 2018 guidelines shall be implemented in EV-related studies in the conetext of nomenclature, processing, separation, characterization, and functional studies of sample EVs. Combined with the ability to sort individual particles and next-generation sequencing, these technologies could open a whole new era for both EV characterization and EV-based diagnosis. Additionally, it has been recently proven that proteomic profiling of EVs can provide novel ways for the prediction of treatment responses, the classification of tumors of unknown origin, and the early detection of tumors [174].

The ability of EVs to mediate intercellular communication and modulate target cell behavior has several potential therapeutic applications. EVs derived from several immune cells, such as DCs and NK cells, have been efficiently tested as potent anti-tumor therapies [175,176]. A striking recent example is the use of CAR T cell-derived EVs as a therapeutic alternative to CAR T therapy. Indeed, it was shown that EVs derived from CAR T cells were able to exert antitumor activity and show higher safety than CAR T cell therapy itself [177]. Recently, a group showed that B cell-derived EVs reduce T cell activation in response to chemotherapy, and therefore inhibiting B cell EV secretion could increase chemotherapy efficacy [178]. The discovery of autologous neoantigens in cancer EVs reveals a high potential for EV-based neoantigen vaccines and therapeutics, without the manipulation of host cells that could trigger unexpected responses. In addition, EVs release inhibitors that have been studied and tested in pharmacological applications [179]. The up-to-date mechanisms and range of EV inhibitors are still rather unspecific and unable to cover all types of EV release. Future work aimed to selectively block the release of detrimental EVs without affecting the ones with physiological roles will allow for new therapeutic approaches to suppress cancer growth and spread. EVs have also been proposed as flexible cargo for therapy delivery, showing great ability to be manipulated and exert the desired effect [180]. Considering all this evidence, EVs are emerging as a potent and flexible therapeutic tool that could be implemented for the diagnosis and treatment of several diseases. Technological development in the coming years will allow us to fully exploit their benefit for early diagnosis and as novel therapeutic options to be used alone or in combination with standard clinical regimens.

## Figures and Tables

**Figure 1 cancers-12-03696-f001:**
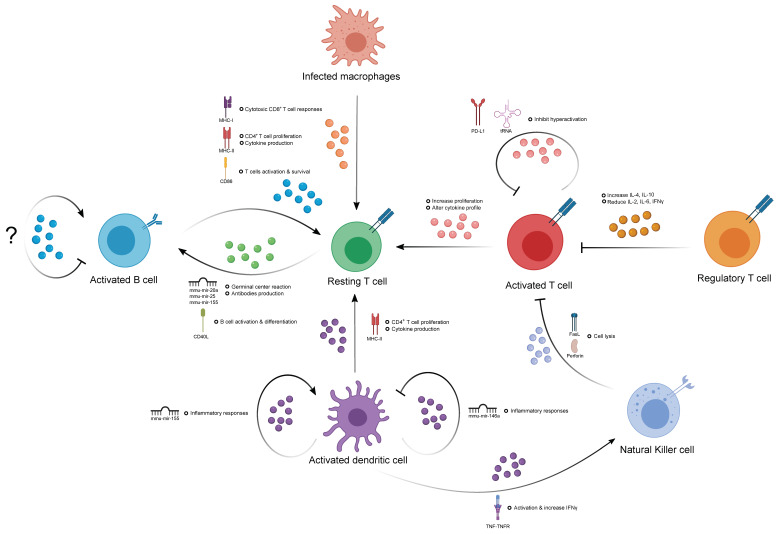
Extracellular vesicle (EV)-mediated immune–immune regulation. This figure depicts the intra- and inter-cellular communications between different immune cells via the exchange or delivery of the specific EV molecular cargo indicated and how these could exert positive or negative regulatory effects in certain immune cells.

**Figure 2 cancers-12-03696-f002:**
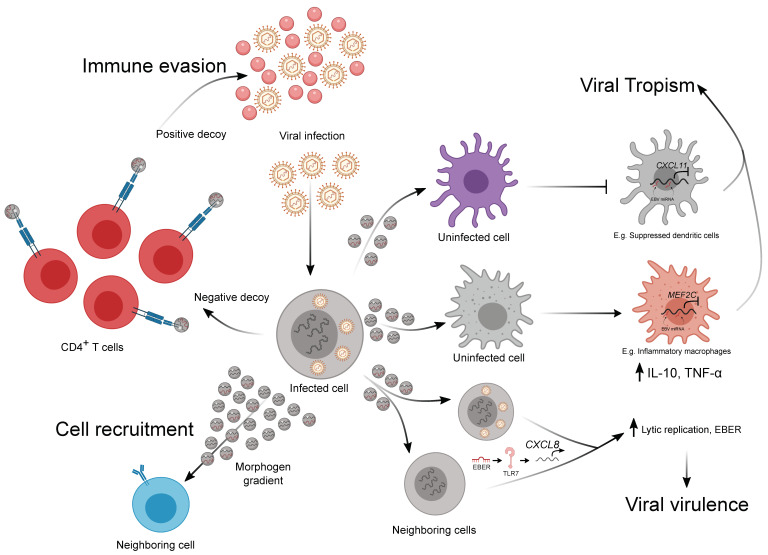
Extracellular vesicle-mediated immune–viral regulation during infection. This figure depicts the potential mechanisms by which virus-encoding EVs could alter virus–host interactions mediated by specific viral EV molecular cargo as indicated. These include the regulation of immune evasion, viral tropism, viral virulence, and potentially immune cell recruitment.

**Figure 3 cancers-12-03696-f003:**
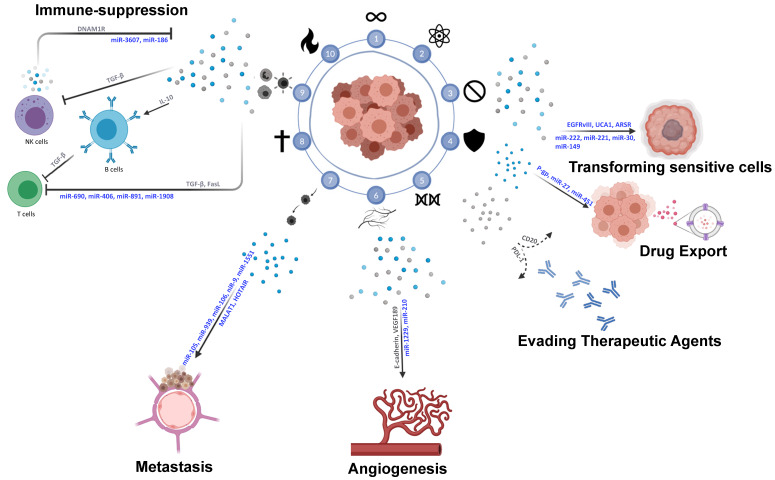
Extracellular vesicle-mediated tumor microenvironment (TME) communication. Schematic depiction of interactions in the TME between the tumor and surrounding cells promoting immune-suppression, angiogenesis, metastasis, and therapy-resistance. Factors and EVs in blue denote uptake-dependent processes; factors and EVs in grey denote uptake-independent processes.

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
