# Peer review of "Extracellular Vesicles Orchestrate Immune and Tumor Interaction Networks"

_cancers, 2020, doi:10.3390/cancers12123696_

Round 1
Reviewer 1 Report
The review manuscript by Yim Wai et al. gives a concise overview of the role of extracellular vesicles as mediators of inter-cellular interaction in the context of immune homeostasis, infection, and cancer. The research focus of the manuscript is relevant and, despite that a lot of work has been done to investigate extracellular vesicles, the implications for disease biology and therapeutic opportunities to target them still remain to be understood. The manuscript reads nicely and covers (potential) mechanisms of action, future prospects for extracellular vesicle biology, and implications for therapeutics. However, there are a few additional questions that should be covered in detail for a comprehensive review of the topic, particularly with the focus on novelty in the field.
Mechanisms of action.
- EVs as a potential source of tumour neo-antigens and anti-tumour immune reactivity
- PD-L1 and co. on extracellular vesicles as an immune evasion mechanism
- The authors mention epigenetics in the introduction, but should also cover the mechanisms of epigenetic regulation by EVs in the different contexts.
- How do normal cells interact with tumour cells through EV-mediated cargo exchange?
- What is the role of fibroblasts, mesenchymal stem cells, and other non-immune stromal cells as important components in the tumour microenvironment according to growing evidence?
- What is the potential role of EVs in a cancer stem cell reversion mechanism?
Therapeutics.
- What are the latest advances and future prospects for using exosome inhibitors for cancer?
- What are the specific challenges that are required for using extracellular vesicles for monitoring disease or diagnosis in the clinics?
- Use of EVs as a source of neoantigens in cancer therapy.
Future prospects of the field. What are the technological advances required to reach the point of high-throughput studies of extracellular vesicles and their potential use in the clinics? How useful are current technologies, such as multiplex imaging flow cytometry for large EV detection? It would be important to underline the technologies that have advanced the field so far and the limitations. What are the challenges for reproducibility of EV content and physical characteristics?
Figures. Figure 1 nicely describes the multiple ways cargo exchange between immune cells can activate or suppress an immune response centered on T cells and B cells. PDL1 expressing proteins should be mentioned as inhibition mechanisms of activated T cells in addition to tRNA exchange.
All figure legends in the submitted manuscript should describe in detail the corresponding figure, even if the information has been provided in the main text. For better readability, the authors should increase the font size specifically in Figure 3.
Author Response
Reviewer 1:
Comment: The review manuscript by Yim Wai et al. gives a concise overview of the role of extracellular vesicles as mediators of inter-cellular interaction in the context of immune homeostasis, infection, and cancer. The research focus of the manuscript is relevant and, despite that a lot of work has been done to investigate extracellular vesicles, the implications for disease biology and therapeutic opportunities to target them still remain to be understood. The manuscript reads nicely and covers (potential) mechanisms of action, future prospects for extracellular vesicle biology, and implications for therapeutics. However, there are a few additional questions that should be covered in detail for a comprehensive review of the topic, particularly with the focus on novelty in the field.
Response: We thank the Reviewer for the appreciation of our work, and we thank her/him for the suggestions that have helped to further improve the quality of the manuscript.
Mechanisms of action.
Response: We included several sections in the manuscript following each of the Reviewer suggestions below
Comment: EVs as a potential source of tumour neo-antigens and anti-tumour immune reactivity
Response: edited (lines 501-503)
Comment: PD-L1 and co. on extracellular vesicles as an immune evasion mechanism
Response: edited (lines 506-508)
Comment: The authors mention epigenetics in the introduction, but should also cover the mechanisms of epigenetic regulation by EVs in the different contexts.
Response: edited (lines 312-315)
Comment: How do normal cells interact with tumour cells through EV-mediated cargo exchange?
Response: edited (lines 169 -172)
Comment: What is the role of fibroblasts, mesenchymal stem cells, and other non-immune stromal cells as important components in the tumour microenvironment according to growing evidence?
Response: edited (lines 365-380)
Comment: What is the potential role of EVs in a cancer stem cell reversion mechanism?
Response: edited (lines 382-394)
Therapeutics.
Comment: What are the latest advances and future prospects for using exosome inhibitors for cancer?
Response: edited (lines 571-575)
Comment: What are the specific challenges that are required for using extracellular vesicles for monitoring disease or diagnosis in the clinics?
Response: edited (lines 529-530)
Comment: Use of EVs as a source of neoantigens in cancer therapy.
Response: edited (lines 569-571)
Comment: Future prospects of the field. What are the technological advances required to reach the point of high-throughput studies of extracellular vesicles and their potential use in the clinics? How useful are current technologies, such as multiplex imaging flow cytometry for large EV detection? It would be important to underline the technologies that have advanced the field so far and the limitations. What are the challenges for reproducibility of EV content and physical characteristics?
Response: edited (lines 531-538)
Comment: Figures. Figure 1 nicely describes the multiple ways cargo exchange between immune cells can activate or suppress an immune response centered on T cells and B cells. PDL1 expressing proteins should be mentioned as inhibition mechanisms of activated T cells in addition to tRNA exchange.
Response: We thank the Reviewer for their suggestion. Indeed, we discussed this in the text but did not report it in the figure. We now included PDL1 as an additional mechanism of T cells inhibition in the new Figure 1
Comment: All figure legends in the submitted manuscript should describe in detail the corresponding figure, even if the information has been provided in the main text. For better readability, the authors should increase the font size specifically in Figure 3.
Response: As the reviewer suggested, we added more details in the figure legends (lines 111-114; 239 to 242) and we improved the quality of Figure 3 to allow better readability.
Reviewer 2 Report
Extracellular vesicles orchestrate immune and tumor interaction networks
EVs are since years important topic in immune and tujor interactions. Yim et al, descriped EV an starting from multiple pathways and biogenesis of EVs to the role of EVs inimmune homeostasis. The figures are very descriptive and contribute to better understanding. Especially the elaboration of the immune cells released eVs is very well done. The In total 156 literature references have been used. The compilation of literature on EVs and the work on the chosen topics of this group is truly remarkable.
However, in my opinion, too little attention was paid to the MISEV criteria in the selection of the references. The detection of EVs is a challenge due to their size, the reproduction of the results often leaves researchers desperate, therefore the MISEV criteria were created to detect EVs better at least by different technologies and controls. In some publications used, the MISEV criteria are missing and the presence of EVs can be questioned. Out of 156 literature references only three of the J. extracellular vesicle were used.
Introducing a point regarding the problem of EV detection instead of future therapeutic perspective for EVs would be more useful.
Minor: 465: two dots
Author Response
Reviewer 2:
Comment: Extracellular vesicles orchestrate immune and tumor interaction networks.
EVs are since years important topic in immune and tumor interactions. Yim et al, described EV an starting from multiple pathways and biogenesis of EVs to the role of EVs in immune homeostasis. The figures are very descriptive and contribute to better understanding. Especially the elaboration of the immune cells released eVs is very well done. The In total 156 literature references have been used. The compilation of literature on EVs and the work on the chosen topics of this group is truly remarkable.
Response: We thank the Reviewer for the positive comments, the appreciation of our figures and the recognition of the value of our work for the community.
Comment: However, in my opinion, too little attention was paid to the MISEV criteria in the selection of the references. The detection of EVs is a challenge due to their size, the reproduction of the results often leaves researchers desperate, therefore the MISEV criteria were created to detect EVs better at least by different technologies and controls. In some publications used, the MISEV criteria are missing and the presence of EVs can be questioned. Out of 156 literature references only three of the J. extracellular vesicle were used.
Response: We apologize for not having mentioned MISEV guidelines, which are indeed a reference for the whole EVs community. In the revised manuscript we provide information on the MISEV criteria (lines 538-540) and included references to relevant papers in JEV (ref number 13, 89, 156, 173, 179).
Comment: Introducing a point regarding the problem of EV detection instead of future therapeutic perspective for EVs would be more useful.
Response: We thank the Reviewer for this suggestion. We now included a statement on the relevant problem of EVs detection (lines 531-534). However, we believe that this perfectly fits in the “Future therapeutic perspectives” section of our manuscript. Solving the problem of EVs detection is indeed the first step for taking EVs to the clinic in terms of biomarkers detection and therapeutic applications.
Comment: Minor: 465: two dots
Response: Amended